# Opioid-Induced Reductions in Amygdala Lateral Paracapsular GABA Neuron Circuit Activity

**DOI:** 10.3390/ijms24031929

**Published:** 2023-01-18

**Authors:** Joakim W. Ronström, Natalie L. Johnson, Stephen T. Jones, Sara J. Werner, Hillary A. Wadsworth, James N. Brundage, Valerie Stolp, Nicholas M. Graziane, Yuval Silberman, Scott C. Steffensen, Jordan T. Yorgason

**Affiliations:** 1Department of Psychology/Neuroscience, Brigham Young University, Provo, UT 84602, USA; 2Department of Pharmacology/Anesthesiology and Perioperative Medicine, Pennsylvania State University College of Medicine, Hershey, PA 17033, USA; 3Department of Neural and Behavioral Sciences, Pennsylvania State University College of Medicine, Hershey, PA 17033, USA; 4Department of Cellular Biology and Physiology, Brigham Young University, Provo, UT 84602, USA

**Keywords:** basolateral amygdala, paracapsular, intercalated, morphine, GABA, opioid

## Abstract

Opioid use and withdrawal evokes behavioral adaptations such as drug seeking and anxiety, though the underlying neurocircuitry changes are unknown. The basolateral amygdala (BLA) regulates these behaviors through principal neuron activation. Excitatory BLA pyramidal neuron activity is controlled by feedforward inhibition provided, in part, by lateral paracapsular (LPC) GABAergic inhibitory neurons, residing along the BLA/external capsule border. LPC neurons express µ-opioid receptors (MORs) and are potential targets of opioids in the etiology of opioid-use disorders and anxiety-like behaviors. Here, we investigated the effects of opioid exposure on LPC neuron activity using immunohistochemical and electrophysiological approaches. We show that LPC neurons, and other nearby BLA GABA and non-GABA neurons, express MORs and δ-opioid receptors. Additionally, DAMGO, a selective MOR agonist, reduced GABA but not glutamate-mediated spontaneous postsynaptic currents in LPC neurons. Furthermore, in LPC neurons, abstinence from repeated morphine-exposure in vivo (10 mg/kg/day, 5 days, 2 days off) decrease the intrinsic membrane excitability, with a ~75% increase in afterhyperpolarization and ~40–50% enhanced adenylyl cyclase-dependent activity in LPC neurons. These data show that MORs in the BLA are a highly sensitive targets for opioid-induced inhibition and that repeated opioid exposure results in impaired LPC neuron excitability.

## 1. Introduction

Opioid use disorder (OUD) has become a rising problem throughout the world, with stress and anxiety being major factors for initial drug seeking and relapse [1,2,3]. In animal models, repeated exposure to the opioid agonist morphine induces long-term increases in anxiety-like behaviors [4]. The basolateral amygdala (BLA) is a critical brain region for the regulation of stress responses and anxiety-like behaviors. Increases in glutamatergic (GLUergic) BLA principal neuron activity results in anxiety-like behavior [5,6], while the inhibition of BLA principal neuron activity decreases anxiety-like behavior [7,8,9]. BLA principal neuron activity is regulated by two groups of inhibitory gamma aminobutyric acid (GABAergic) neurons: local interneuron populations are distributed throughout the BLA proper and dense clusters of feed-forward GABAergic neurons are located along the BLA/external capsule border, termed the lateral paracapsular (LPC) neurons or lateral intercalated cells [10,11,12,13].

While relatively little is known about LPC neurons, the activation of LPC inputs to BLA principal neurons has been shown to reduce anxiety-like behavior [14]. Furthermore, the inactivation of these cells results in impaired fear extinction [15]. In contrast to other BLA neurons, LPC neurons highly express inhibitory μ-opioid receptors (MORs) [16], causing LPC neurons to be a potential target of opioids that may become dysregulated following chronic exposure. Such LPC neuron dysregulation may be critical for enhanced BLA excitation and subsequent increases in anxiety-like behaviors that may drive opioid seeking and relapse. Therefore, understanding the mechanisms by which acute and repeated MOR activation modulate LPC neuron activity may provide important insights in the development of amygdala-related dysfunction and perhaps dysregulation of anxiety-reward circuits. 

MORs and adenosine A1 receptors (A_1_R) belong to a class of G-protein-coupled receptors (GPCR) that activate inhibitory G_i_ proteins [17,18]. The acute activation of MORs inhibits adenylyl cyclase, resulting in reduced cyclic adenosine monophosphate (cAMP) levels [19] and reduced protein kinase A (PKA) activity [20], which ultimately inhibits neuronal activity. In contrast, long-term opioid use and withdrawal is associated with alterations in adenylyl cyclase function [21,22,23,24]. In the ventral tegmental area (VTA), chronic morphine-induced cAMP-dependent increases the probability of GABA release onto VTA dopamine neurons [25]. In addition to the obvious similarities in signaling between opioid and adenosine receptors, adenosine signaling is implicated in many different aspects of opioid withdrawal behavior, though the interactions are complex and much about brain region specific changes is unknown [26]. Therefore, the hypothesis of the present study is that LPC neurons are sensitive to acute opioid exposure and opioid-induced adaptations in LPC neuron circuit activity, including adenosine/cAMP signaling sensitivity and potentially MOR-dependent anxiety-related behavior. 

## 2. Results

### 2.1. Visualization of GABAergic LPC Neurons of the BLA

Widefield and confocal images were taken of the amygdala sections from the GAD65-mCherry/GAD67-GFP transgenic mice, demonstrating fluorescent identification of GABAergic LPC neurons (Figure 1A–E). A widefield confocal image of the BLA is depicted in Figure 1A–C, without a confocal aperture in order to reduce optical sectioning and show diffuse fiber bundles. Note the dense clustering of GAD65/67 positive neurons along the dorsal and lateral BLA region indicating LPC neurons. The GAD fluorescent labeling for both reporters was generally observed in all the LPC neurons. Similar neuronal populations are also found in the medial paracapsular (MPC) area and in the main island (MI) of the amygdala. Local GABA interneurons were also observed in this preparation. The LPC clusters appeared packed with dense GABA terminal labeling, as seen with diffuse labeling in confocal sections (Figure 1D). This is despite observing dense cell bodies in these areas when an optically sectioning aperture is unused for widefield fluorescent imaging (Figure 1C). Similarly, during patch clamp recordings, the LPC neurons were identified using infrared Dodt imaging superimposed on GAD-dependent epifluorescence on a widefield microscope, with diffuse but intense labeling around cell bodies (Figure 1E). 

Previous studies have demonstrated opioid receptor expression in LPC neurons [27,28]. Similarly, dense MOR antibody co-labeling was observed in GAD67-GFP positive LPC neurons (Figure 2A–C) and, throughout the BLA, which is similar to that reported previously [29,30]. Delta opioid receptor (DOR) labeling was also observed in LPC, MPC, and BLA neurons (Figure 2D–F). In Figure 2E, differential interference contrast imaging was used to observe the unlabeled cells and blood vessels (shown as dark edges) and respective DOR labeling (overlay in Figure 2F). MOR and DOR labeling occurred throughout the BLA, including on LPC neurons. Thus, MOR and DOR receptors are opioid receptor targets on LPC neurons. 

### 2.2. Acute MOR Activation Reduces LPC Neuron Excitability

In order to establish a functional role for MORs in LPC signaling, whole cell recordings were performed in LPC neurons and the acute effects of the selective MOR agonist [D-Ala2, N-Me-Phe4, Gly5-ol]-enkephalin (DAMGO; 10 µM) were tested. The LPC neurons are thought to form highly connected intra-LPC neuron networks that modulate GABA release [31]. Therefore, spontaneous inhibitory postsynaptic currents (sIPSC) measurements may serve as an indirect measure of ongoing neighboring LPC neuron activity but may reflect GABA activity from other nearby GABA neurons (LPCs, BLA, or lateral amygdala). sIPSC were observed in LPC neurons and DAMGO reduced sIPSC frequency by ~46%, which was reversed by the non-selective antagonist naloxone (Figure 3A,B; 1 µM; one-way ANOVA repeated measures, *F*_(2,12)_ = 7.074, *p* = 0.0093, with Tukey’s post hoc analysis, baseline vs. DAMGO, *p* < 0.01, baseline vs. Naloxone and DAMGO vs. Naloxone, *p* > 0.05). The sIPSC amplitude appeared unaltered by DAMGO and Naloxone (Figure 3C; *p* > 0.05). In contrast to the effects on sIPSCs, the spontaneous excitatory postsynaptic currents (sEPSCs) appeared unaffected by DAMGO (Figure 3D,E; *p* > 0.05). Since the sIPSC frequency was reduced, but not altogether silenced, by opioid application, the sIPSCs appear to come from opioid sensitive and insensitive sources. The GABAergic LPC expression of MORs indicates that the inhibition of sIPSCs by DAMGO is likely due to changes in release from local LPC neurons.

### 2.3. Morphine Withdrawal Reduces LPC Neuron Excitability

Next, the effects of morphine exposure and withdrawal on LPC neuron activity were examined. The mice received either saline or morphine (10 mg/kg) IP injections for five consecutive days, followed by a two-day withdrawal period (experimental timeline: Figure 4A). We next examined the LPC neuron response to an injected current, a measure of intrinsic membrane excitability (IME), since the IME sets the action potential threshold and controls the firing frequency (Figure 4B). In morphine-treated mice, the AP amplitude (54.76 ± 0.91 mV) was greater than in saline-treated mice (273 APs from 12 total cells; 46.16 ± 1.1 mV; two-tailed, unpaired *t*-test; *t*_(271)_ = 5.878, *p* < 0.0001; Figure 4C). The resting membrane potential was similar across conditions (two-tailed, unpaired *t*-test; *t*_(8)_ = 1.719, *p* = 0.1239; Figure 4D). The number of APs from the current injection was greater in the saline-treated mice (two-way ANOVA repeated measures; time: *F*_(5,40)_ = 9.315, *p* < 0.0001; drug: *F*_(1,44)_ = 5.119, *p* = 0.050; interaction: *F*_(5,40)_ = 2.446, *p* = 0.0481; Figure 4E) and the AP variability was reduced in the morphine-treated mice (two-tailed unpaired *t*-test; *t*_(11)_ = 2.297, *p* < 0.0422; Figure 4F). We next investigated the AHP, which is critically involved in regulating the frequency and timing of action potentials, with decreases in AHP resulting in increases in neuronal firing rates and increases in AHP resulting in decreases in the neuronal firing rates. The AHP was greater in the morphine-treated mice (−9.99 ± 0.406 mV × msec to −5.588 ± 0.391 mV × msec; two-tailed, unpaired *t*-test; *t*_(271)_ = 7.78, *p* < 0.0001; Figure 4G), and morphine-treated mice had shorter AP durations (1.587 ± 0.056 msec to 1.227 ± 0.058 msec; two-tailed, unpaired *t*-test; *t*_(271)_ = 4.42, *p* < 0.0001; Figure 4H). Overall, the LPC neurons of morphine exposed mice are less excitable than saline controls, suggesting differences in signaling for the two conditions.

### 2.4. Multiple Morphine Exposures Increase Adenylyl Cyclase and Adenosine A_1_R Activity in LPC Neurons

Acutely, opioids decrease adenylyl cyclase activity through the activation of G_i_ coupled signal transduction mechanisms [32,33,34,35]. However, prolonged exposure increases adenylyl cyclase (likely as an adaption to suppressed activity), resulting in increased cAMP sensitivity during acute morphine withdrawal [23,32]. We sought to determine if this adaptation is present in LPC neurons after morphine (Figure 5A–C). Since intracellular phosphodiesterases (PDEs) break down cAMP, PDE inhibitors can be used to uncover underlying changes in adenylyl cyclase activity [36,37]. The bath application of the PDE inhibitor isobutyl methylxanthine (IBMX; 100 µM) had no effect on the sIPSC amplitude between saline-treated or morphine-treated mice (saline: 27.27 ± 2.869 pA to 28.25 ± 3.013 pA; morphine: 22.50 ± 0.676 pA to 21.30 ± 1.614 pA; unpaired two-tailed *t*-test: *t*_(20)_ = 1.153, *p* = 0.263, Figure 5B). IBMX increased the sIPSC frequency in morphine-treated mice by ~35% (0.5449 ± 0.1282 Hz to 0.7175 ± 0.165 Hz) (Figure 5A,C), which was significantly different from the IBMX effects in the saline controls, which were reduced (~7% reduction; 0.3017 ± 0.078 Hz to 0.2948 ± 0.065 Hz; unpaired two-tailed *t*-test: *t*_(11)_ = 3.779, *p* = 0.003, Figure 5C). Thus, inhibiting PDE activity reveals a difference in cAMP-related signaling pathways in morphine-treated mice (i.e., increased cAMP). 

Since PDE-cAMP activity appears enhanced by prolonged morphine exposure [19,20], other targets sensitive to changes in adenosine tone may also be affected. Therefore, the effects of morphine exposure on G_i_ signaling in the non-opioid, adenosine mediated GPCR A_1_R were tested [38] (Figure 5D–F). The sIPSC amplitude was unaltered by A_1_R activation (N6CPA, 1 µM) but was reduced with A_1_R inactivation (DPCPX, 300 nM) in both animal groups (Figure 5E; two-way repeated measures ANOVA; main effect N6CPA/DPCPX *F*_(2,66)_ = 11.33, *p* < 0.001; main effect morphine: *F*_(1,67)_ = 2.64, *p* = 0.119; interaction: F_(2,66)_ = 1.44, *p* = 0.249). This effect was slightly, but not significantly, lower (~15%) in the morphine- compared to saline-treated mice. A_1_R activation reduced the frequency (Figure 5F) in both saline- (2.692 ± 0.7815 Hz to 1.833 ± 0.4712 Hz) and morphine- (0.9005 ± 0.1849 Hz to 0.4523 ± 0.0889 Hz) treated mice, which was reversed by DPCPX in saline- (~24% increase) and greater in morphine- (~121% increase) treated mice (0.4523 ± 0.0889 Hz to 1.119 ± 0.2499 Hz; two-way repeated measures ANOVA; main effect N6CPA/DPCPX: *F*_(2,52)_ = 27.57, *p* < 0.001, main effect morphine: *F_(1,52)_* = 9.1, *p* = 0.007; interaction: *F*_(2,52)_ = 16.33, *p* < 0.001). The increased activity observed in the morphine-treated mice after A_1_R inactivation suggests an increased adenosine tone in these animals. A_1_R activation-induced decreases in sIPSC frequency are probably due to the inhibition of LPC neurons through the inactivation of the cAMP cascade. The morphine-withdrawn mice have been shown to upregulate the adenylyl cyclase pathway and antagonize the A_1_R with DPCPX activating this pathway even further in the nucleus accumbens [39]. Thus, DPCPX increased the synaptic activity in the LPCs, which may be through increasing cAMP levels (since A_1_R is Gi coupled). However, increased activity could also be due to the DPCPX-mediated activation of silent synapses.

## 3. Discussion

The BLA is implicated in motivational behavior including reinforcement learning, decision-making, drug seeking, and stress-induced drug seeking (for review see [10]). LPC neurons are a major source of feed-forward inhibition onto principal BLA neurons [10] and participate in the regulation of anxiety-like behaviors [14]. Therefore, it is crucial to understand how opioids alter LPC activity and subsequent impacts on neurocircuitry and expressed behavior. The present study examines amygdala circuit function and morphine-induced adaptations in LPC circuit activity using patch–clamp electrophysiology. LPC neurons express opioid receptors as confirmed via MOR and DOR antibody labeling. Whole-cell electrophysiology experiments in LPC neurons demonstrated opioid sensitive GABAergic but not GLUergic inputs. Morphine exposure and withdrawal result in reduced intrinsic excitatory activity and increased adenylyl cyclase activity on GABA inputs to LPC neurons. These data highlight the importance of opioid receptors in LPC circuit specific adaptations during morphine exposure and withdrawal, which may have implications in craving and stress-related behavior.

### 3.1. Opioid Receptor Effects in the BLA

Opioid receptors couple to inhibitory effectors that reduce adenylyl cyclase activity, resulting in lower cAMP-dependent signaling [18,19]. GABAergic LPC neurons express MORs and DORs, suggesting that both receptors are involved in morphine-induced decreases in LPC excitability. We chose to focus on LPC MORs because of the known high expression and potential behaviors such as stress-induced analgesia and adaptation to stress [30]. However, the activation of LPC DORs also has behavioral implications that show opposing results [40] but similar effects as MOR activation on projections onto amygdala paracapsular neurons [27]. Reduced BLA MOR expression attenuates opioid-induced behavioral changes [40]; therefore, MORs could be a major contributor to opioid-induced cellular adaptations in this region. MOR activation decreases LPC activity and naloxone reversed this effect. The selective inhibition of LPC neurons can activate BLA pyramidal neurons through disinhibitory processes [10,41], which may underlie associated anxiety-like behaviors [42]. Although BLA network activity is regulated by GABAergic LPC neurons, the precise mechanisms, including mechanisms of plasticity are still largely unknown. Further work will be needed to elucidate the precise role of LPC neurons in controlling these pathways and subsequent behavioral outcomes.

Morphine is a powerful reinforcer, producing conditioned place preference in mice (CPP) that extinguishes in the absence of morphine and reinstates with stressors [43,44]. Morphine exposure enhances GLUergic signaling between the BLA and NAc [45], which may underlie stress-induced drug-seeking behaviors [46,47]. Morphine exposure and withdrawal produce anxiety-like behavior in rodents [48,49]. The underlying cellular mechanisms that drive these behaviors have strong implications for the BLA and further hodological studies will need to be performed to uncover the specific subcircuits that are involved in these behaviors.

Acutely, DAMGO decreased the frequency of sIPSCs while the sEPSC frequency and amplitude were unaffected. The inhibition of sIPSCs was reversed by naloxone, providing strong supporting evidence that the opioid receptors are heavily involved in the LPC-BLA circuitry [27,41]. MOR and DOR agonism has been shown to reduce IPSCs in amygdala intercalated cells [27]. The present study expands on this work by focusing on LPC neurons specifically, demonstrating MOR and DOR expression in LPCs and the functional inhibition of IPSCs, likely through the inhibition of collateral inputs. Furthermore, DAMGO’s effects were specific to changes in sIPSC frequency, suggesting presynaptic changes in upstream (other LPCs, lateral amygdala, etc.) neuronal excitability.

Prolonged morphine exposure resulted in decreased excitability in LPC neurons, as depicted in current-injection-mediated depolarization experiments. Since the intrinsic properties (AP Bursting, AHP, and AP duration) of LPC neurons were reduced after morphine withdrawal, this suggests that LPC neuronal ion channel function or channel expression are fundamentally changed. The increase in AP amplitude indicates increased NaV conductance and the larger AHP indicates increased KV conductance. The specific voltage-gated channels underlying the excitability in LPC neurons are not functionally characterized; however, several gene candidates have been identified for voltage-gated sodium (SCN1A) and voltage-gated potassium channels (KCNH5, KCNS2, KCND2, and KCNAB1) in intercalated amygdala neurons [50], suggesting that these channels may be a source of morphine-induced adaptations in LPC neuron excitability. Thus, changes in AP frequency and intrinsic AP properties appear to be through increases in NaV and KV channel function/expression during morphine withdrawal, which would also manifest as increases in membrane conductance.

### 3.2. Morphine Effects on the LPC Adenosine System

The activation of cAMP-dependent processes through IBMX application increased the sIPSC frequency in LPCs of morphine-treated animals, indicating increased GABA release. In general, opioid withdrawal produces increases in adenylyl cyclase-initiated cellular mechanisms [20], which is consistent with the present findings. Increased GABA release observed herein after PDE and A_1_R inhibition is supportive of our hypothesis of morphine-induced adaptions in LPC PKA activity. PKA is widely expressed throughout the brain and is an effective modulator of neuronal targets that influence cellular excitability such as ion channels [51]. Therefore, this kinase represents a potential target for morphine-induced adaptations.

The A_1_R acts as an autoreceptor for adenosine [52]. Similar to DORs and MORs, it is G_i_-coupled and its activation is associated with increased K+ channel activity [53], decreases in voltage-gated Ca^2+^ channel activity [54], lower cAMP levels from adenylyl cyclase inhibition, and subsequently reduced PKA [52,55,56]. The neuronal levels of adenosine, the primary endogenous agonist of A_1_Rs, result from exonucleotide interactions with ATP, causing adenosine receptor activity to be a measure of extracellular ATP tone [52]. As such, adenosine and related receptors serve as significant modulators of neural activity by affecting the release of other neurotransmitters (e.g., GLU, GABA, acetylcholine, and dopamine; [57]). Opioid withdrawal increases adenosine and A_1_R activation in various regions of the brain, resulting in reduced neuronal activity [58,59]. Increased adenosine extracellular tone and cellular A_1_R activity are associated with the negative side effects of long-term opioid use for pain management [60,61,62,63]. A_1_R sensitization is also associated with chronic opioid use and withdrawal [64,65]. Presently, IBMX application increased sIPSCs in LPC neurons, likely through an increase in cAMP (via phosphodiesterase inhibition) and downstream effects on enhancing local GABA release onto LPC neurons [52,55,56,64]. Since IBMX and DPCPX both have increased effects in morphine-exposed mice and LPC neurons express opioid receptors, increases in adenylyl cyclase and A_1_R activity may be directly within local LPC neurons, though cellular specific expression of A_1_R in LPC neurons will need to be confirmed. The downstream effects of adenylyl cyclase on ion channels in morphine-treated mice likely include enhanced voltage-gated K^+^ and Na^+^ channel function [53,66,67] and may be responsible for changes in AP kinetics observed herein.

Small conductance Ca^2+^-activated K^+^ (SK1-3) channels are one potential target of PKA. These channels are shown to regulate neuronal firing frequency in various parts of the central nervous system by contributing to the AHP of an action potential [68,69]. SK channels are phosphorylated targets of PKA and the activation of PKA results in a decrease in SK2 channels in the plasma membrane of COS7 cells [70]. Since Gi GPCRs reduce adenylyl cyclase and PKA activity, morphine should acutely reduce PKA activity, resulting in increased SK expression. In this study, LPC neurons in morphine-treated mice exhibited strengthened AHP, which may be indicative of greater SK expression, though the specific channels underlying this effect will need to be tested explicitly.

Voltage-gated calcium channels are also chief players in initiating and regulating neurotransmitter release. A large body of research has shown that PKA activity affects L-type calcium channels, an interaction that has vast consequences on cardiac function [71]. It is less known, however, the role PKA may have on P/Q- and N-type calcium channels, which are channels known to mediate synaptic transmission [72]. Other kinases, such as protein kinase C, enhance N-type calcium conductance and contribute to enhanced neurotransmitter release [73]. Although not directly shown, it is possible that PKA activation may have a similar effect in regulating neuronal activity in the present morphine model.

## 4. Materials and Methods

### 4.1. Animals

Female heterozygous glutamate decarboxylase (GAD) 67-green fluorescent protein (GFP) knock-in mice [74] were bred with male homozygous GAD65-mCherry knock-in mice (JAX#023140) [75] to create double-labeled mice on a mixed CD-1XC57/BL6 background. The mice were provided with ad libitum access to food and water and maintained on a 12:12 h light/dark cycle. For repeated morphine exposure, mice were injected intraperitoneally (IP) with morphine (10 mg/kg/day) or an equivolume of saline (~0.3–0.4 mL) for 5 consecutive days with a sterile needle. The animals were returned to their home cages immediately following injection and observed for a Straub tail response [76,77]. On days 6–7, all the mice received IP saline, which is described herein as a from 24 to 48 h morphine withdrawal period. The mice were then euthanized and used in electrophysiology experiments. All the protocols and animal care procedures were in accordance with the National Institutes of Health Guide for the care and use of laboratory animals and approved by the Brigham Young University IACUC. 

### 4.2. Brain Slice Preparation and Drug Application 

Isoflurane (Patterson Veterinary)-anesthetized mice (30–45 days old; male and female) were euthanized by rapid decapitation and the brains were immediately removed, sectioned into 220 μm-thick coronal slices (Leica VT1000S, Vashaw Scientific, Roswell, GA, USA), and incubated for 60 min at 34 °C in preoxygenated (95% O_2_/5% CO_2_) aCSF consisting of (in mM): 126 NaCl, 2.5 KCl, 1.2 NaH_2_PO_4_, 1.2 MgCl_2_, 21.4 NaHCO_3_, and 11 d-glucose. The cutting solution also contained either the NMDA GLU antagonist MK801 (10 µM; (5S,10R)-(+)-5-methyl-10,11-dihydro-5H-dibenzo[a,d]cyclohepten-5,10-imine; Abcam, Cambridge, UK) or the GLU antagonist kynurenic acid (2 mM) for the blockade of ionotropic GLU receptors. After incubation, the tissue was transferred to aCSF (34 °C) with kynurenic acid (2 mM) for recording the GABA currents (equilibrium potential approximately −40 mV) through sIPSCs or picrotoxin (100 µM) for recording sEPSCs. Kynurenic acid was used as a non-selective glutamate receptor blocker [78,79,80]. The following concentrations of drugs (acquired from either Sigma-Aldrich or Tocris Bioscience) were both applied for slice electrophysiology experiments where specified: the MOR agonist DAMGO (10 μM), the nonselective opioid receptor antagonist naloxone (1 μM), the cAMP phosphodiesterase inhibitor IBMX (100 μM), the adenosine A_1_R agonist N6CPA (1 μM), or the A_1_R antagonist DPCPX (300 nM).

### 4.3. Histology, Immunohistochemistry and Microscopy

The mice were anesthetized with isoflurane (CHEBI:6015; Patterson Veterinary, Loveland, CO, USA), transcardially perfused with 4% paraformaldehyde (1× PBS solution). After dissection, the brains were post-fixed overnight in the same fixative at 4 °C and cryoprotected in 30% sucrose dissolved in PBS for an additional 48 h at 4 °C. The coronal cryosections were obtained and left free-floating for labeling. The sections were permeabilized in PBS with 1% Triton X-100 (Thermo Fisher Scientific, Waltham, MA, USA, 9002-93-1), 5% normal goat serum (NGS; Aurion, 900.077), and 1% Fc block (BD Biosciences, San Jose, CA, USA 553142) for 1 h at room temperature with slight agitation. For primary antibody exposure, the brain sections were incubated overnight at 4 °C with slight agitation in PBS with either rabbit anti-OPRM1 (1:300, Novus Biologicals, NB100-1620, RRID: AB_10003258) or rabbit anti-OPRD1 (1:300, LS Bio, LS-C381221) with 0.1% Triton, 5% NGS, and 1% Fc block. For secondary antibody exposure, the sections were rinsed two times with PBS for 10 min each and then incubated for 2 h and 30 min at room temperature in PBS with either Alexa Fluor 594 donkey anti-rabbit (1:300, abcam, ab150076, RRID: AB_2782993) or Alexa Fluor 488 goat anti-rabbit (1:500, Thermo Fisher Scientific, A-11008, RRID: AB_143165) with 0.1% Triton, 5% NGS, and 1% Fc block. The immunostained sections were then mounted onto glass slides and put on coverslips with a Fluoromount g mounting medium (SouthernBiotech). The confocal images were collected using an Olympus IX81 microscope with the FV1000 Fluoview system (Olympus) with UApo/340 20×/0.75 and PlanApo 60×/1.40 objectives (Olympus). The widefield epifluorescent and brightfield transmitted images were acquired using an inverted Nikon Eclipse FN-1 and upright Olympus BX-51WI.

### 4.4. Electrophysiology Recordings

The LPC neurons were identified based on anatomic location and fluorescence. During the experiments, the slices were continually perfused with warm (37 °C) aerated (95% O_2_/5% CO_2_) aCSF at a rate of 2 mL/min. The GABA neurons were identified by the presence of soluble GFP (GAD67) or mCherry (GAD65) fluorescence using an Olympus BX51 upright microscope with Micromanager 1.4 software and an Infinity 3S camera (Lumenera, Ottawa, ON, Canada). The whole cell recordings were performed from the LPC region of the amygdala using a Multiclamp 700 B amplifier (Molecular Devices, Sunnyvale, CA, USA) at a holding potential of −60 mV. The glass pipettes (2.5–6 MΩ; PP-83 Micropipette Puller; Narishige, London, UK) were filled with a K-Gluconate pipette solutions (in mM: 110 K-gluconate, 10 KCl, 15 NaCl, 1.5 MgCl_2_, 10 HEPES, 0.1 EGTA, 2 Mg-ATP, 0.2 Na-GTP, 10 Na2-Phosphocreatine tetrahydrate, and 5 QX-314; pH 7.3–7.4, 278 mOsm). The series resistance was monitored and maintained below approximately 15 MΩ for inclusion in the analysis. The drugs were applied for 4–6 min. The current clamp recordings used current step pulses of −20 pA and 60 pA. The voltage clamp data were filtered at the amplifier with a 4-pole high-pass Bessel filter at 2–10 kHz. The recordings were sampled continuously at 5 kHz with an Axon Digidata 1440 A (Molecular Devices) and at 10 kHz using pClamp 10 (Molecular Devices, San Jose, CA, USA) or Axograph 10 (Axograph, Sydney, Australia). 

### 4.5. Statistical Analysis

The electrophysiology recordings were analyzed in Clampfit 10 (Molecular Devices) and Mini Analysis (Synaptosoft, Decatur, GA, USA). The spontaneous synaptic current rates for baseline and post-drug conditions were measured across a 2 min period. The statistics were performed using Prism 5 (GraphPad) and NCSS 8. The statistical significance was determined for groups of 2 variables using a two-tailed Student’s *t*-test. The experiments with >2 groups with only one factor were tested for significance using a one-way analysis of variance (ANOVA). When data were from multiple time points from the same experiment, a repeated-measures ANOVA was performed. For experiments that examined multiple factors and possible interactions, two-way ANOVA or repeated-measures ANOVAs were used. Tukey’s HSD and Bonferroni correction methods were used for post hoc analysis. The action potential (AP) characteristics were measured using Clampfit 10 and intrinsic properties were examined by analyzing the resting membrane potential, instantaneous firing frequency, integrated area under curve, and AP duration. The half-width was calculated as the duration (in sec) for the AP at half height. After, the hyperpolarization area (AHP) under the curve was calculated as the negative area during the repolarization stage in the AP. Unpaired *t*-tests were used to determine statistical significance.

## 5. Conclusions

Acute MOR activation reduced LPC neuron firing and sIPSC frequency. Morphine withdrawal reduced LPC neuron excitability. Together, these experiments demonstrate the significant inhibitory effect that MOR activation has on LPC neurons. With decreased LPC activity, we would predict a corresponding increase in BLA activity [10,12,14,81] associated with increases in anxiety-like behaviors [82] and the reinstatement of use preference involved in substance use disorders [83]. The current data show that morphine exposure increases adenylyl cyclase-dependent activity in LPC neurons and produces adenosine A_1_R sensitivity to antagonism, suggesting additional receptor tone. While A_1_R activation decreased sIPSC activity acutely, adaptations resulted in decreased LPC excitability during acute withdrawal and enhanced A_1_R responsivity to antagonism. These adaptations appear to include intrinsic changes, such as voltage-gated ion channels, resulting in changes in membrane resistance and action potential conductance, as well as adenylyl cyclase-dependent cascading effects. G_i_ binding to adenylyl cyclase, whether as a result of A_1_R or MOR activation, causes inhibition of the cAMP pathway, resulting in decreased sIPSCs in LPC neurons, and adaptations in cAMP signaling to oppose the inhibition. The present work demonstrates a novel mechanism by which LPC neuron function is regulated, thereby affecting the function and regulation of BLA pyramidal neurons. However, future studies are needed to assess behavioral implications of LPC circuitry adaptations. Overall, these studies indicate that behavioral and pharmacological interventions targeting the LPC neuronal circuitry may prove to have potential beneficial treatment options for substance use disorders and potentially other mental health issues mediated by this circuit including stress and anxiety.

## Figures and Tables

**Figure 1 ijms-24-01929-f001:**
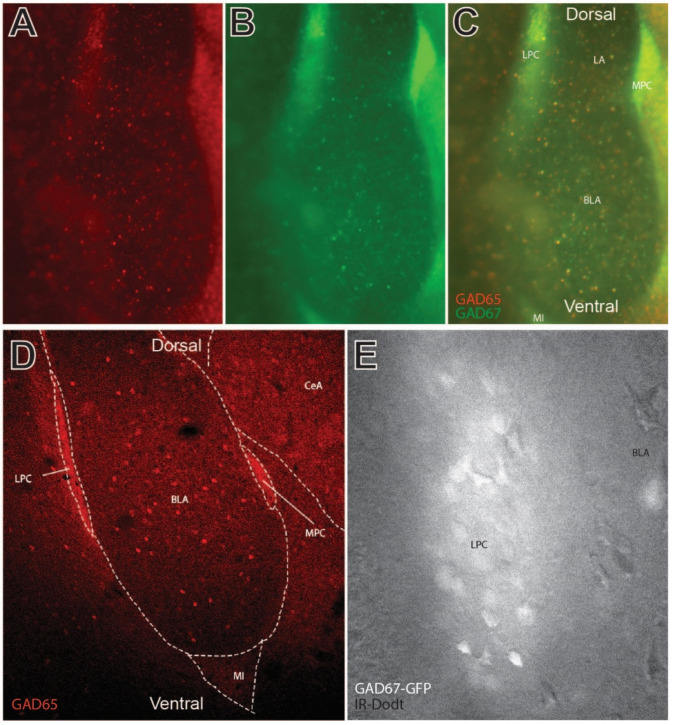
Visualization of GABAergic lateral paracapsular (LPC) neurons of the basolateral amygdala (BLA) in coronal sections. (**A**–**C**) Widefield fluorescent imaging of GAD65 (mcherry) and GAD67 (GFP)-labeled GABAergic neurons in and around the BLA, lateral amygdala (LA), and central amygdala (CeA). Paracapsular neurons of this region include GABAergic neurons of the LPC, medial paracapsular (MPC), and main island (MI) subregions. GAD65 and GAD67 varied to some degree by cell, though strong colocalization was observed in LPC and MPC neurons. (**D**) Confocal sectional imaging in GAD65 (mcherry)-labeled transgenic mice shows GAD65 labeling of axon collaterals in LPC and MPC clusters. (**E**) High magnification infrared image of the dense LPC neuron cluster as visualized on a patch electrophysiology microscope. Note the strong fluorescent labeling of individual cells and diffuse fluorescence reminiscent of axon collaterals observed in panel D. Fluorescent GFP labeling of GABAergic neurons was used to direct recordings in LPC electrophysiology experiments.

**Figure 2 ijms-24-01929-f002:**
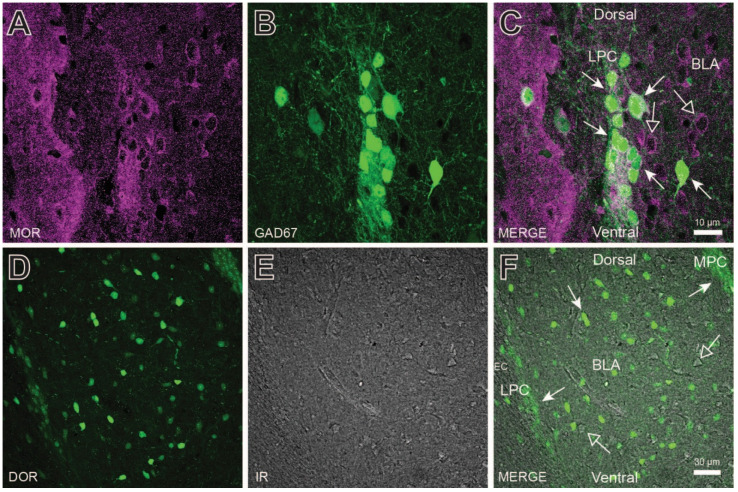
Opioid receptor expression in lateral paracapsular (LPC) neurons and throughout the basolateral amygdala (BLA). (**A**–**C**) mu-opioid receptor (MOR) antibody labeling of the LPC cluster and BLA in GABAergic (GAD67GFP+, closed arrows) and non-GABAergic (GAD67GFP-, open arrows) neurons. (**D**–**F**) Delta-opioid receptor (DOR) antibody labeling of the LPC and MPC clusters and BLA. DOR positive cells (closed arrows) were observed throughout the BLA, but are also surrounded by DOR negative cells (open arrows) as visualized using IR. Although MOR and DOR antibody labeling were concentrated in LPC and MPC clusters, both labels were observed to some degree throughout the BLA.

**Figure 3 ijms-24-01929-f003:**
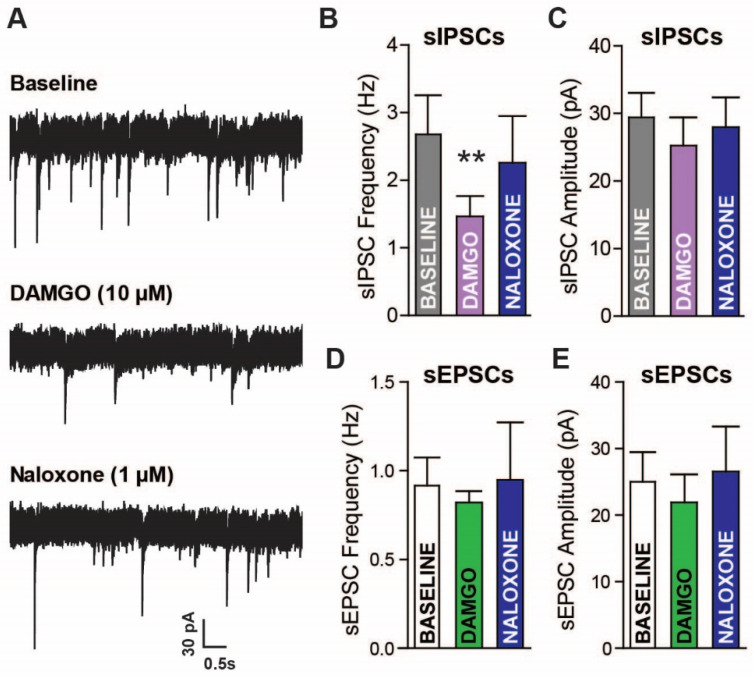
Acute mu-opioid (MOR) activation reduces GABAergic but not glutamatergic currents in amygdala lateral paracapsular (LPC) neurons. (**A**) raw spontaneous inhibitory postsynaptic currents (sIPSCs) recorded during artificial cerebrospinal fluid (aCSF) (baseline), DAMGO (10 µM), and naloxone (1 µM) application. Whole cell voltage-clamp electrophysiology was used to measure the frequency (**B**) and amplitude (**C**) of currents in LPC neurons. sIPSC frequency was reduced by MOR activation, which reversed with MOR antagonism. sEPSCs were also recorded in LPC neurons, with no clear effects of MOR activation/inactivation on sEPSC (**D**) frequency and (**E**) amplitude. Baseline vs. DAMGO ** *p* < 0.01.

**Figure 4 ijms-24-01929-f004:**
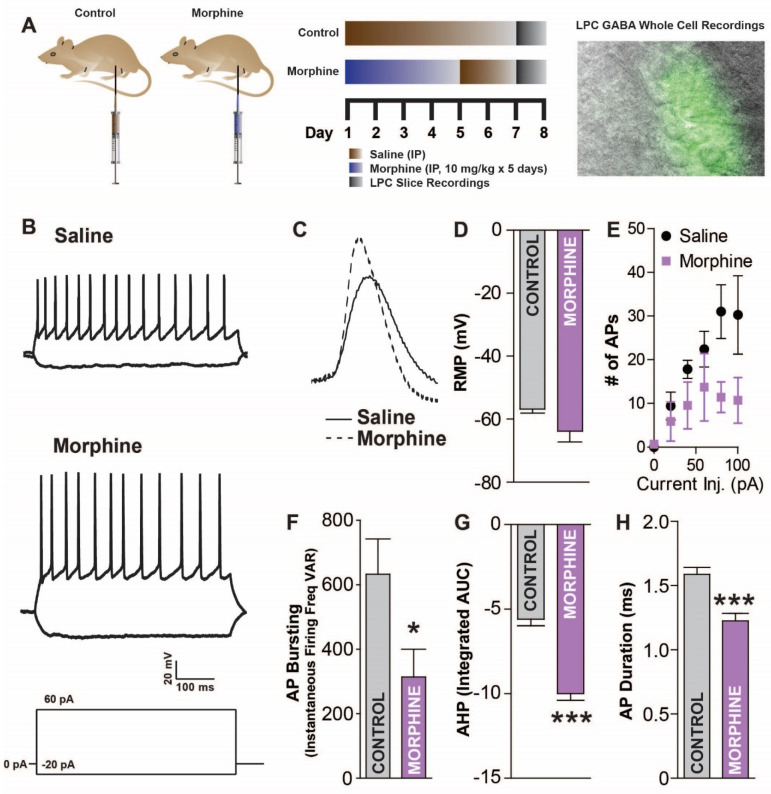
Acute morphine withdrawal reduces lateral paracapsular (LPC) GABAergic neuron excitability. (**A**) Saline or morphine (10 mg/kg) injected mice (IP; days 1–5 morphine/saline, days 6–7 saline), treated as shown in the experimental timeline (top right) were examined for changes in LPC function. Basolateral amygdala (BLA) slices were obtained (day 8) and LPC GABAergic neurons recorded from using whole-cell current-clamp electrophysiology during IR and fluorescent widefield imaging (top right). (**B**) Raw traces of current injection induced depolarizations in LPC neurons for saline- or morphine-treated mice. (**C**) Example action potential (AP) traces in LPC GABA neurons from saline and morphine mice. (**D**) Resting membrane potential is slightly but not significantly lower in morphine mice. (**E**) Number of APs from a depolarizing current (500 ms) is reduced in morphine withdrawal. (**F**) Variability in instantaneous firing frequency during current injection was reduced in morphine withdrawn mice. (**G**) AP afterhyperpolarization (AHP) area under the curve (AUC) is increased in morphine mice. (**H**) AP duration is reduced in morphine withdrawn mice. * *p* < 0.05, *** *p* < 0.001.

**Figure 5 ijms-24-01929-f005:**
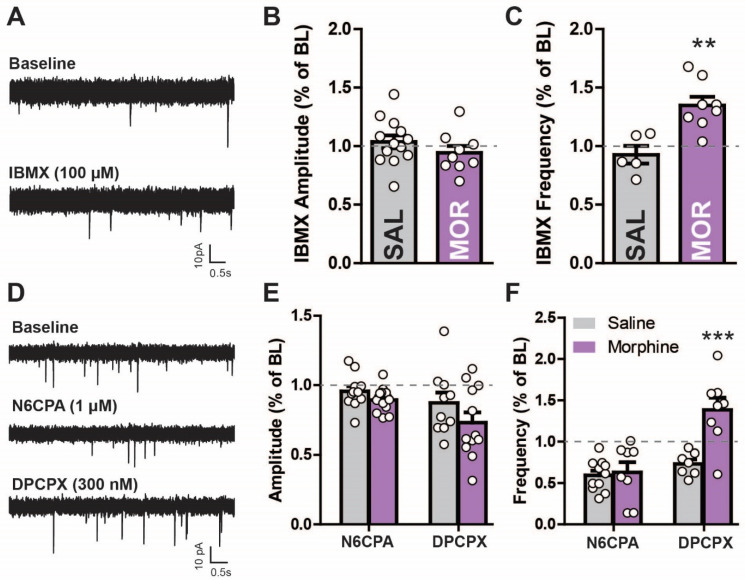
Morphine withdrawal increases cyclic AMP-dependent activity in lateral paracapsular (LPC) neurons. (**A**) Raw current traces from voltage clamp recordings in GABAergic LPC neurons measuring sIPSCs in saline- (IP, 7 days) and morphine-treated mice (10 mg/kg for 5 days, saline 2 days) in aCSF (baseline) and in the presence of the phosphodiesterase inhibitor IBMX (100 µM) measuring sIPSC (**B**) amplitude and (**C**) frequency. IBMX increased frequency more in morphine- than saline-treated mice. (**D**) Raw current tracings of sIPSCs from voltage clamp recordings in GABAergic lateral paracapsular (LPC) neurons in saline- (IP, 7 days) and morphine-treated mice (10 mg/kg for 5 days, saline 2 days). The selective A_1_R agonist N^6^-cyclopentyladenosine (N6CPA; 1 µM) was applied, followed by reversal with the antagonist 8-cyclopentyl-1,3-dipropylxanthine (DPCPX; 300 nM) while measuring sIPSC (**E**) amplitude and (**F**) frequency. sIPSC frequency increased to a greater extent after DPCPX in morphine- compared to saline-treated mice. ** *p* < 0.01, *** *p* < 0.001.

## Data Availability

The data presented in the study are included in the article. Further inquiries can be directed to the corresponding authors.

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
