# Peer review of "Opioid-Induced Reductions in Amygdala Lateral Paracapsular GABA Neuron Circuit Activity"

_ijms, 2023, doi:10.3390/ijms24031929_

Round 1

Reviewer 1 Report

Could increased contributions of glutamatergic receptors not affected by Kynurenic acid contribute to increased spontaneous EPSCs by driving LPCs? This is based on my understanding that Kynurenic acid primarily targets NMDA receptors and has a concentration-dependent effect on AMPARs.  , Understandably, finding the source of the non-opioid/GABAR effects is not the primary goal of these studies it might be a source of the insensitive component.  

The quality of several photomicrographs appeared to be on the low end making details hard to visualize. 

Data that were presented are all in withdrawal animals. Are there any comparisons/data to animals that have not undergone withdrawal? As withdrawal from any number of substances alters synaptic signaling it would be helpful to understand if these changes are withdrawal dependent. 

Very clearly written manuscript and presentation of data.

Reviewer 2 Report

The current work is interesting and sound relevant. Indeed, not so many work deals with intercalated cell masses. The manuscript is descriptive, but any scientific knowledge has to be first demonstrated. However the discussion is very (too) large and does not focus enough on the explanation of the observed results and correlations, and more specifically on the paradoxical results of Fig 5F.

The first part of the work is pretty straightforward. However the transition to Adenosine signaling needs to be better justified. Is there any known relationship between Adenosine signaling and modulation of GABA synaptic transmission during addiction (or in control condition), apart from the fact that MOR and A1R share the same intracellular pathway. The justification has to be improved.

The second part of the experiment deals with withdrawal, therefore groups should not be named morphine. Indeed, no experiment were performed after repetitive morphine injections but after the consequence of its withdrawal. Even (see below) if this state as to be justified.

I have organized my comments in four sections; methodological, then comments concerning the results, discussion and conclusion. The methodological concerns have to be answered. The remaining comments could be answered by rephrasing or complementary experiment/analysis or supported by references.

Methodological concerns

Nowadays, it is often mandatory to experiment on male and female. Is this mandatory for IJMS? If not, authors have to justified their choice of testing their hypothesis on female only.

Working with IPSC, Chloride equilibrium potential value, should be given in M&M.

Spike analysis is not described in M&M, how AP duration (half-width) was measured? How AHP AUC was set?

Withdrawal protocol is not described in M&M. Indeed, a 5 days injection of constant morphine dose (10 mg/kg) is sometime not enough, due to tolerance, to induce morphine addiction. Was addiction checked or was the protocol supported by references? Sometime withdrawal is also enhanced by naloxone injection.

Quality of the photo in the pdf file is poor, photos are blurry especially in panel 1B, where staining seems unspecific because no soma could be guessed.

Fig2E is not readable

Comments on results

The conclusion drawn on Fig1D in text (dense GABA terminal labeling around cell bodies), cannot be confirmed by  looking at the figure, cell bodies are not visible in LPC or MPC.

MOR was not observed in BLA? whereas McDonald and Neugebauer have! Moreover Fig 2C show MOR+GAD+ cell in BLA, therefore the conclusion in lines 175-176 should take that in account, instead of letting think that only LPC express MOR.

Line 187, what is the meaning of the general region in fig2 legend, does it mean BLA?

line 194, where are located the nearby GABA neurons, BLA? LPC?

Fig 3, were kinetics also affected by DAMGO? Absence of kinetics alteration would be an additional proof of synapse silencing.

AP amplitude is also dependent on RMP, Fig 4 shows that RMP is more negative in morphine group even if not significantly it may be enough to affect AP amplitude.

Morphine increase excitability (number of AP) but if membrane resistance is altered by morphine treatment, this could explain the higher excitability. Is there any difference in membrane resistance between morphine and saline groups?

In figure 4 and throughout the manuscript, Instead of morphine, withdrawal should be a more accurate/appropriate name of the group.

Figure 4 illustrates the result of a fixed depolarizing step. But the amplitude of the step is dependent on the membrane resistance, therefore a better measurement should be the exact value of the membrane potential reached during the step, which would provide a controlled, stable and reproducible experiment.

Lines 264-268 do not clearly commented IBMX effect, which “increase is greater in morphine group than in saline”. However, in saline IBMX induces a decrease of sIPSC frequency whereas it's an increase in morphine group. This sentence has to be rephrased.

Line 270, the transition is weak. Previous experiment showed an increase in sIPSC frequency, but there is no measurement of PDE-cAMP activity only a correlation. What is the (known) link between cAMP and synaptic transmission?

Effect of A1R inactivation effect speaks for a tonic effect!

How could authors explain that A1R agonist effect on frequency was reversed by antagonist in saline, but that antagonist per se seems to have an effect, because, it reverses agonist effect and also increases the frequency. The explanation provides line 284-286 is weak, and needs to be strengthen by experiments or supported by published papers.

Activation of A1R reduced synaptic transmission frequency but not amplitude meaning that some synapse are silenced but intrinsic synaptic  properties are not modified. what is known at that level?

However a more intriguing effect is the further blockade of A1R that not only reverse the consequence of the activation but in morphine group increase sIPSC frequency to a level above control meaning an increase of synaptic activity or the activation of silent synapse in LPC or in upstream structure. This should be better explained and discussed.

Comment on discussion

Line 314, is there any published proof about withdrawal and fear-related behavior?

Which is the LPC upstream structure?  How does it fit in the present scheme?

Line 345 OR = opioid receptor?

Lines 355 and following, an alternative explanation that could explain part or the entire changes induced by morphine is a modification membrane resistance, that will produce the same consequences on AP frequency and amplitude. Which could also in fine be due to channel function or expression alterations.

Line 368-369, “A1R inhibition is indicative of morphine induced adaptions in LPC PKA activity” is too speculative!

Line 382-383, what is the link between A1R and morphine withdrawal?

Comments on the conclusion

Line 419-420, no conclusion can be drawn on prolonged morphine treatment because it was not tested in this paper restricted to withdrawal only.

Line 427, adenosine A1R sensitivity demonstrates by  N6CPA is identical in control and withdrawal group (except when antagonized by DPCPX), therefore the sensitivity cannot be linked to morphine exposure!

Sentence lines 425-428 seems central to the conclusion, but is to complex and should be rephrased may be in two or three sentences.

Could authors clarify "adaptation" line 428 and 430!

Line 433-436 is correct but how a treatment could be specific to LPC when the same receptors and pathways are present in BLA and CeA (neighboring nucleus of the same brain structure).

Reviewer 3 Report

The present study was designed to test the hypothesis that opioid exposure changes the excitability of lateral paracapsular (LPC) 19 GABAergic inhibitory neurons that reside along the BLA/external capsule border using immunohistochemical and electrophysiological approaches. The results showed that LPC neurons and other nearby GABA and non-GABA neurons expressed MORs and δ-opioid receptors. Additionally, DAMGO, a selective MOR agonist, reduced GABA- but not glutamate-mediated spontaneous postsynaptic currents in LPC neurons. Furthermore, in LPC neurons, abstinence from repeated morphine exposure in vivo (10 mg/kg/day, 5 days, 2 days off) resulted in significant decreases in intrinsic membrane excitability, a ~75% increase in afterhyperpolarization, and a ~40-50% increase adenylyl cyclase-dependent activity in LPC neurons. The authors concluded that MORs in the BLA are highly sensitive targets for opioid-induced inhibition and that repeated opioid exposure results in impairments in LPC neuron excitability. This is a well designed study that replicates neuroadaptations that have been observed with opioid drugs in other brain circuits that involve GABAergic synapses but breaks new ground in focusing on the LPC of the BLA. There are a few design issues that should be addressed.

1.     The sex of the animals from which the cells were recorded is not specified. It would be interesting to see those data. Assuming there were cells from both males and females, it would be useful to test for sex differences.

2.     The authors evoke a further hypothesis in the Abstract that such LPC activity participates in neuroadaptations that contribute to anxiety-like behavior during opioid withdrawal, but no causal relationship is tested. No behavioral studies are included that show that inactivation of the cells under study here actually contribute to anxiety-like behavior that is associated with opioid withdrawal. The authors argue that hodological studies are needed, but this reviewer would like to see some causality studies. Given the lack of an actual causal link between the LPC and opioid withdrawal-induced anxiety, the authors’ conclusion, “Overall, these studies indicate that pharmacological interventions targeting LPC neurons may prove to be beneficial treatment options for substance use disorders and potentially other mental health issues mediated by this circuit including stress and anxiety,” seems a bit overstated.

3.     The chronic opioid-induced inhibition of GABAergic neurons in the ventral tegmental area have been well studied and follow a similar pattern of disinhibition and rebound withdrawal-induced inhibition. For example, chronic morphine induced a cAMP-dependent increase in the probability of GABA release onto VTA dopamine neurons (Madhavan et al., 2010). A few sentences that relate changes in these two brain areas may be of interest.

Madhavan A, He L, Stuber GD, Bonci A, Whistler JL. micro-Opioid receptor endocytosis prevents adaptations in ventral tegmental area GABA transmission induced during naloxone-precipitated morphine withdrawal. J Neurosci 2010;30(9):3276e86.

4.     I applaud the authors for carefully reviewing the literature, but I would like to mention this very early paper that reported that opioid withdrawal increased adenylate cyclase:

Collier, H. O. & Francis, D. L. Morphine abstinence is associated with increased brain cyclic AMP. Nature 255, 159–162 (1975). 

Round 2

Reviewer 2 Report

Authors have answered to my comments and amended the manuscript accordingly (when needed).